# Cytogenetically Cryptic Acute Promyelocytic Leukemia: A Diagnostic Challenge

**DOI:** 10.3390/ijms241713075

**Published:** 2023-08-22

**Authors:** Maedeh Mohebnasab, Peng Li, Bo Hong, Jennifer Dunlap, Elie Traer, Guang Fan, Richard D. Press, Stephen R. Moore, Wei Xie

**Affiliations:** 1Department of Pathology and Laboratory Medicine, Oregon Health and Science University, Portland, OR 97239, USA; 2Division of Molecular Genomic Pathology, University of Pittsburgh Medical Center, Pittsburgh, PA 15213, USA; 3Department of Molecular and Medical Genetics and Knight Diagnostics Laboratory, Oregon Health and Science University, Portland, OR 97239, USA; 4Division of Hematopathology, Department of Pathology, University of Utah Health, Salt Lake City, UT 84112, USA; 5Division of Hematology and Medical Oncology, Oregon Health and Science University, Portland, OR 97239, USA

**Keywords:** APL, *PML*::*RARA*, cryptic translocation, FISH

## Abstract

Cytogenetically cryptic acute promyelocytic leukemia (APL) is rare, characterized by typical clinical and morphological features, but lacks t(15;17)(q24;q21)/*PML::RARA* translocation seen in conventional karyotyping or FISH. The prompt diagnosis and treatment of APL are critical due to life-threatening complications associated with this disease. However, cryptic APL cases remain a diagnostic challenge that could mislead the appropriate treatment. We describe four cryptic APL cases and review reported cases in the literature. Reverse transcriptase polymerase chain reaction (RT-PCR) is the most efficient diagnostic modality to detect these cases, and alternative methods are also discussed. This study highlights the importance of using parallel testing methods to diagnose cryptic APL cases accurately and effectively.

## 1. Introduction

Acute promyelocytic leukemia (APL) is a unique entity and accounts for 10–15% of newly diagnosed acute myeloid leukemia (AML) [1,2]. APL patients are at high risk for disseminated intravascular coagulopathy (DIC) and associated life-threatening hemorrhagic conditions. Early diagnosis is critical for immediate treatment with all-trans-retinoic acid (ATRA) and arsenic (ATO) to improve outcomes [1,3]. More than 95% of APL cases harbor reciprocal translocation t(15;17)(q24;q21), leading to *PML::RARA* fusion, visualized by karyotype [1,3]. This translocation results in myeloid precursor maturation arrest in the promyelocytic stage, and promotes cell growth by blocking apoptotic cell death [4]. There is also a minor subset of cases with APL morphologic features, which harbors variant translocations in which *RARA* is fused to a partner gene other than *PML* [1,5].

Traditionally, conventional chromosome analysis and fluorescence in situ hybridization (FISH) are required to detect *PML::RARA* fusion to confirm the diagnosis of APL [5]. However, in rare cases, patients with clinical and morphological features of APL lack a detectable *PML::RARA* fusion by conventional chromosomal analysis or FISH. Cytogenetically cryptic APL accounts for less than 1% of APL cases [6]. In this study, we have characterized four cryptic APL cases with classic morphologic and clinical findings, but t(15;17)/*PML::RARA* fusion was not revealed by conventional cytogenetic analysis or interphase FISH. Subsequently, reverse transcription polymerase chain reaction (RT-PCR) confirmed *PML::RARA* fusion in all cases. We further reviewed 55 reported cryptic APL cases in the literature and summarized the morphologic and immunophenotypic features of these cases and the possible underlying mechanisms that make alternative methods necessary for an accurate and prompt diagnosis.

## 2. Case Report

Patient 1: A 49-year-old man with a history of Factor V. Leiden and multiple deep vein thrombosis presented with fever, fatigue, and headache. His peripheral blood (PB) showed pancytopenia with rare atypical blasts/promyelocytes. Flow cytometric analysis of the PB detected 1.5% blasts/promyelocytes, which were positive for CD13, CD33, CD64, CD117, and MPO. These atypical blasts lacked expression of CD34, HLA-DR, and CD11b. The bone marrow (BM) was hypercellular with 90% atypical blasts/promyelocytes (Figure 1). A 220-gene next-generation sequencing (NGS) panel detected mutations in *WT1* and *HNRNPK*. The patient was treated with ATRA and ATO, and remained alive at 26-months follow-up.

Patient 2: A 20-year-old man presented with fatigue, petechiae, subdural hemorrhage, and laboratory findings suggestive of DIC. The PB smear showed 70% characteristic circulating blasts/promyelocytes, with irregular or bilobed nuclei, azurophilic granules, and occasional Auer rods. Flow cytometric analysis performed on the PB showed blasts/promyelocytes to be positive for CD9, CD13, CD33, dim CD56, dimCD64, dimCD117, and MPO, and lacking CD34 and HLA-DR. The patient was treated with ATRA, IDA, and cytarabine and was alive at 24-months follow-up.

Patient 3: A 27-year-old man presented with imaging findings of marrow heterogeneity in T12 during L4–L5 disk herniation work-up. This led to a BM biopsy, which was morphologically concerning for APL. The flow cytometric analysis of the BM detected a blasts/promyelocytes population, positive for CD10, CD11b, CD13, CD16, CD33, CD64, and MPO, and negative for CD34 and HLA-DR. The patient was treated with ATRA and ATO, and is alive.

Patient 4: A 32-year-old woman presented with several weeks of fatigue and easy bruising. She was found to have severe thrombocytopenia and DIC at admission. The flow cytometric analysis of the BM detected blasts/promyelocytes population to be positive for CD2, CD13, CD33, CD117, and MPO. The patient was treated with ATRA and ATO, and was alive at the follow-up.

The demographic features of the four patients are summarized in Table 1. A diagnosis of APL was highly suspected in all four cases initially due to typical morphologic findings (Figure 1A). However, the initial interphase FISH studies using a dual-color dual-fusion probe (MetaSystems, Heidelberg, Germany), which were prompted by the morphologic findings with quick results in 24 h, were negative for *PML::RARA* fusion signal patterns (Figure 1B). The characteristic clinical and morphologic manifestations warranted further evaluation. Subsequently, *PML::RARA* quantitative reverse transcriptase PCR (qRT-PCR) using consensus PCR primers for *PML* and *RARA* detected *PML::RARA* transcript bcr1 breakpoint in cases #1 and #3; bcr3 breakpoint in cases #2 and #4. The conventional chromosomal analysis showed normal karyotypes in three cases at the diagnosis except case #2 with trisomy 8. All four patients were treated with ATRA-based regimen and achieved long-term complete remission (Table 1).

## 3. Discussion

Cryptic APL is a rare event that was first described by Hiorns et al. [7]. It accounts for less than 1% of all APL cases and is characterized by typical morphologic and immunophenotypic abnormalities with unremarkable findings by karyotype and FISH studies [8]. In our study, all patients with cryptic APL were relatively young, and in the low-risk group (white blood cell count less than 10 K/µL) at initial presentation. By extensively searching, we also found 55 cryptic APL cases reported in the literature (Appendix A) involving 20 men and 35 women with a median age of 42.5 years (range: 10–68 years old), similar to the typical APL cases (median age: 47 years old) [9,10]. These cases have classic APL features, with normal banding patterns in chromosomes 15 and 17, through conventional cytogenetics. *PML::RARA* fusion probe was used in forty-one cases, and *RARA* break apart (BA) probe was used in four cases; both fusion and BA probes were used in two cases; and customer-designed probes were used in eight cases. By FISH studies, 39/55 patients were negative for *PML::RARA* fusion, and eight cases showed abnormal signals such as insertion of small part of *RARA* in the *PML* gene, but lacking the typical fusion patterns. Eight cases proved positive by customer-designed probes such as ICRF *PML* and *RARA* cosmid probes. *PML::RARA* fusion was detected in all cases via RT-PCR. Thirty-one patients demonstrated normal karyotype, and 24 patients showed abnormal karyotype without t(15;17) translocation. The most frequent cytogenetic alterations were trisomy 8 and iso(17q). There was no APL variant, where *RARA* fusion occurred with genes different from *PML.* The majority (43 of 46 patients with known therapy) of these patients were treated with ATRA-based regimens. Among 42 patients with follow-up information, 35 were alive, and 7 were deceased due to late-stage disease. The outcomes of the patients with cryptic APL would be similar to those of classic cytogenetically overt APL cases if they were diagnosed and treated promptly, in contrast with APL variant cases, which are associated with adverse outcomes despite treatment with ATRA [10].

FISH is a valuable method to detect typical translocation t(15;17), and was shown to be able to detect *PML::RARA* in 86–90% APL cases [6], with resolution as high as 200 kb, depending on the probe size [11]. However, cryptic APL may be due to submicroscopic insertion that is too small to be hybridized with conventional probes, or producing a subtle signal that is difficult to be visualized using fluorescent microscopy [12,13]. To overcome this challenge, smaller custom probes may be helpful to detect small insertions [12,14]. RT-PCR is used to confirm cryptic APL cases in our study and cryptic cases in the literature, which can detect the common long, variant, and short forms of fusions [15]. While the *RARA* breakpoints occur mainly in intron 2, the *PML* gene has various breakpoints, of which the most common are intron 6 (bcr1), exon 6 (bcr2), and intron 3 (bcr3). Traditionally, these transcript isoforms are classified based on the breakpoint cluster regions (bcrs) of PML and/or the length of the transcript as bcr1 or long, bcr2 or variable, and bcr3 or short transcripts. In our study, two cases were bcr1, and two cases were bcr3. Of 55 cases reported with cryptic APL, 27 (49%) patients were positive for bcr1 (median age 42.2, range 10–68), 2 (4%) cases with bcr2, and 22 (40%) cases with bcr3 (median age 37.8, range 12–68). Previous studies showed no significant difference in age distribution of different isoforms [16,17].

Beyond RT-PCR, other methods such as whole-genome sequencing (WGS), RNA sequencing, Agena fusion panel (Agena MassARRAY, Cerritos, CA, USA), Nanostring gene fusion assay (Nanostring nCounter^®^ Vantage 3D™ Gene fusion Assays, Seattle, WA, USA), and optical genome mapping (OGM) can also detect cryptic APL cases [18,19]. WGS provides comprehensive and unbiased information on gene fusion as well as other structural changes. However, it requires a large amount of sequencing and it is time-consuming and costly [18]. RNA sequencing (e.g., NGS-based targeted RNA fusion panels) is the widely used method in clinical laboratories. In general, sequencing-based methods are vastly affected by library preparation techniques (i.e., amplicon based vs. hybrid capture) and analysis software. While hybrid capture library preparation has a higher sensitivity, it is limited to known fusion partners. Sequencing analysis software also have different sensitivity and specificity to detect and call fusions. As a result, recommendations are to utilize two software packages, simultaneously, to detect fusions [18].

On the other hand, Agena fusion panel requires cDNA synthesis followed by amplification, labeling and spectrometry. This technology utilizes the expression differences in two exons from a particular gene, and contemplates the overexpression of the kinase domain as a result of an activating fusion. A confirmatory next step is performed to detect the fusion partner. Nanostring gene fusion assay applies a combination of hybridization probes and reporter oligonucleotides to capture and label the RNA molecule. The positive signal is produced when the two probes hybridize close enough on the same transcript. The technology comes with an analytical software for data quality and normalization. However, there is not an interpretation pipeline, nor a set cut-off for detection signal. As opposed to Agena MassArray, Nanostring gene fusion assay does not require cDNA synthesis and nucleic acid amplification, and is thus less prone to fixation and PCR artifact [20]. These two technologies are valuable, as they are applicable to FFPE samples with high sensitivity compared to conventional karyotyping and FISH, have a faster turnaround time and a lower cost when compared to sequencing. Nonetheless, these techniques are not suitable for cases with novel fusion partners.

As a new non-sequencing platform, OGM can detect different types of structural variants (SVs) in one assay. In this technique, ultralong high-molecular-weight (ULHMW) DNA molecules are labeled with fluorophore tags specific to different parts of genome and the label pattern is mapped to reference genome and analyzed to reveal structural variants [19]. With >300× coverage and 10% analytical sensitivity, this assay can detect a variety of SVs, including copy number variation, gene level, and exon level insertions (such as internal tandem duplication), inversions, balanced and unbalanced intrachromosomal and/or interchromosomal and three-way rearrangements, with a lower cost and faster turnaround time [19]. However, the assay requires specific extraction techniques and cannot utilize previously extracted DNA, lacks a base level resolution, has regionally low coverage in highly repeated regions like centromeres and telomeres, and cannot provide cell level information (to support sub-clonal alterations) [11].

The limitation of this study is that other alternative methods, such as RNA sequencing, have not been performed in these cases to further confirm the submicroscopic insertion of *RARA* gene into *PML* gene or a possible complex translocation.

In conclusion, cryptic APL is infrequent and while highly curable, it remains a diagnostic challenge. Conventional cytogenetics and FISH studies alone may fail to detect the cryptic *PML::RARA* fusion, which can delay appropriate management. Therefore, we propose in those cases which are highly suspicious for APL, alternative and parallel methods are critical to reach a prompt and accurate diagnosis to improve patient care (Figure 1C).

## Figures and Tables

**Figure 1 ijms-24-13075-f001:**
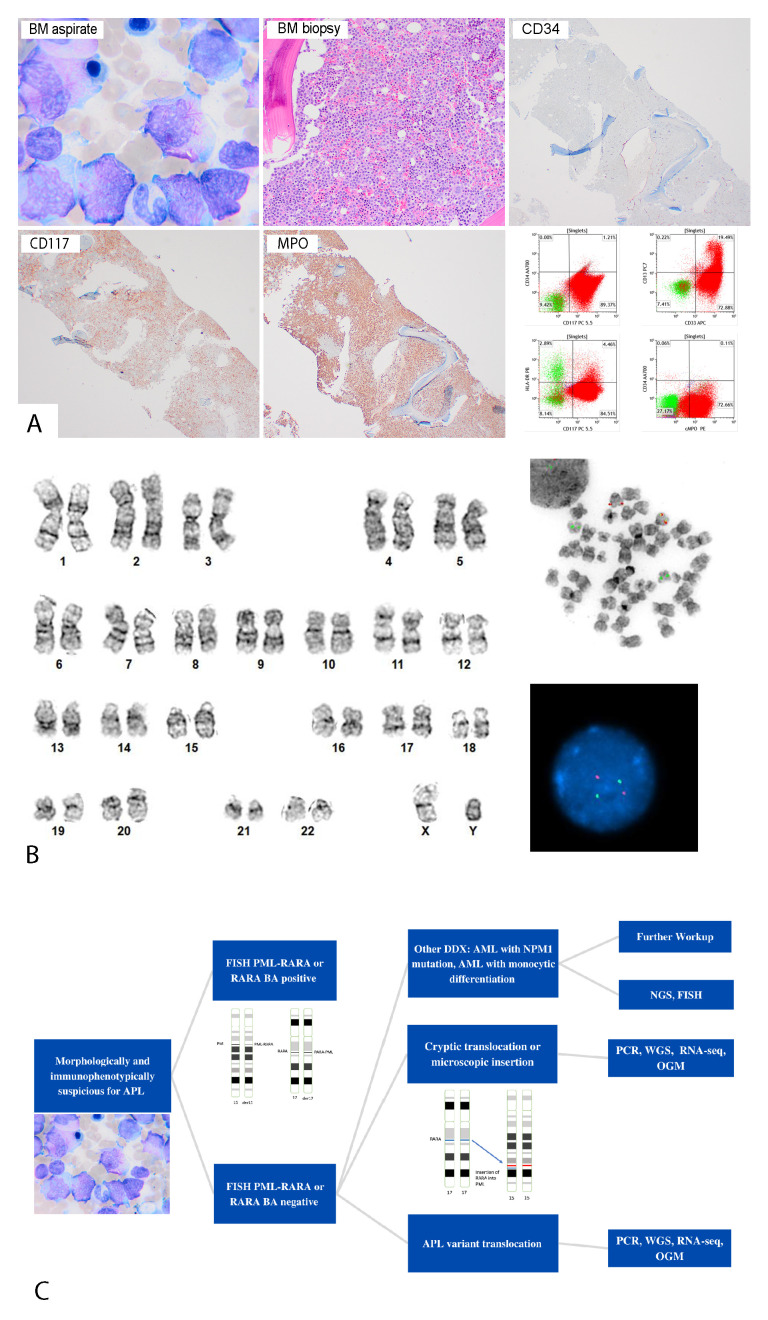
(**A**) Morphologic and immunophenotypic features of a patient (patient #1) with cryptic APL. Bone marrow aspirate smear (upper left panel) (1000×) showed many blasts/promyelocytes with Auer rods. Bone marrow biopsy (upper middle panel) (400×) showed sheets of blasts/promyelocytes negative for CD34, and positive for CD117 and MPO via immunohistochemistry. Flow cytometric analysis (lower right panel) demonstrated that blasts/promyelocytes were positive for CD13, CD33, CD117, and cMPO, and negative for CD34 and HLA-DR. (**B**) Cytogenetic findings of a patient (patient #1) with cryptic APL. Left side: conventional cytogenetic analysis showed a normal male karyotype (46,XY). Right side: FISH analysis was negative for *PML::RARA* fusion in interphase and metaphase. (**C**) Flowchart of diagnosing cryptic APL, morphologic mimics of APL, and *RARA* variant translocation. Abbreviation: DDX, differential diagnosis; FISH, fluorescence in situ hybridization; NGS, next-generation sequencing; OGM, optical genome mapping; PCR, polymerase chain reaction; RNA-seq, RNA sequencing; WGS, whole-genome sequencing.

**Table 1 ijms-24-13075-t001:** Clinical and pathologic features of four patients with cryptic APL.

	Patient 1	Patient 2	Patient 3	Patient 4
Age/Gender	49/M	20/M	27/M	32/F
WBC (×10^9^/L)	0.55	5.91	4.67	1.57
RBC (×10^12^/L)	2.89	5.36	4.69	2.65
Hemoglobin (g/dL)	9.1	16.1	15	8.1
Platelets (×10^9^/L)	51	19	144	68
Circulating blasts (%)	1.5	70	2	0
aPTT	30	29.3	32	33
PT	NA	15.2	13.9	15.4
Fibrinogen	262	<50	325	141
D-dimer	>4	NA	0.5	38
Blasts immunophenotype	Positive for CD9, CD13, dim CD15, CD33, CD38, dim CD45, dim CD64, CD117, MPO	Positive for CD9, CD13, CD33, dim CD56, dimCD64, dimCD117, MPO	Positive for CD10, CD11b, CD13, CD16, CD33, and CD64	CD13, CD33, CD117, bright MPO, and partial CD2 (40%)
Karyotype	46,XY[20]	46,XY[3]/47,XY, +8[17]	46,XY[20]	46,XX[20]
FISH	.ish der(15)t(15;17)(q24;q21)(PML+, RARA+;RARA+)[3]	.ish der(15)t(15;17)(q24;q21)(PML+,RARA+;RARA+)[6]	Negative for *PML::RARA*	Negative for *PML::RARA*
PCR	*PML::RARA* fusion RNA detected	*PML::RARA* fusion RNA detected	*PML::RARA* fusion RNA detected	*PML::RARA* fusion RNA detected
PCR breakpoint	bcr1	bcr3	bcr1	bcr3
220-gene NGS	*WT1, HNRNPK*	ND	ND	ND
Treatment	ATRA + ATO	ATRA + IDA/Cytarabine	ATRA + ATO	ATRA + ATO
Outcome	Alive	Alive	Alive	Alive

Abbreviation: APL, acute promyelocytic leukemia; aPTT, activated partial thromboplastin time; ATRA, all-trans retinoic acid; ATO, arsenic trioxide; bcr, breakpoint cluster region; F, female; FISH, fluorescence in situ hybridization; M, male; NA, not available; ND, not done; NGS, next-generation sequencing; PCR, polymerase chain reaction; PT, prothrombin time; RBC, red blood cell; WBC, white blood cell.

## Data Availability

No new data were created or analyzed in this study. Data sharing is not applicable to this article.

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
