# Peer review of "Cytogenetically Cryptic Acute Promyelocytic Leukemia: A Diagnostic Challenge"

_ijms, 2023, doi:10.3390/ijms241713075_

Round 1

Reviewer 1 Report

This is an excellent paper. I only have 2 minor corrections. 

1. How can you explain the relative younger APL age of diagnosis for your 4 patients presented? Was this co-incidential? Even though the median age of 42.5 years for the 55 cryptic APL cases is similar to the expected overall median 47 years of age, i was wondering about the possibility that bcr1 or/and bcr3 cases might present at an earlier age with milder clinical course.....Was that investigated? Are there other reported APL cases with bcr1 or bcr3 breakpoints and what was the age of diagnosis in such cases? Please comment in the discussion. 

2. In the cryptic APL cohort you studied (55 cases) were there any APL variant cases or not? Please answer that to the discussion. 

Author Response

  1. How can you explain the relative younger APL age of diagnosis for your 4 patients presented? Was this co-incidential? Even though the median age of 42.5 years for the 55 cryptic APL cases is similar to the expected overall median 47 years of age, i was wondering about the possibility that bcr1 or/and bcr3 cases might present at an earlier age with milder clinical course.....Was that investigated? Are there other reported APL cases with bcr1 or bcr3 breakpoints and what was the age of diagnosis in such cases? Please comment in the discussion. Extensive literature search was done, there is no significant difference in age distribution of different isoforms. 

2. In the cryptic APL cohort you studied (55 cases) were there any APL variant cases or not? Please answer that to the discussion. The variant APL case were already removed from the series and all 55 cases were cryptic APL. 

Both of these comments are addressed in discussion section. 

Reviewer 2 Report

The Cryptic APL is determined by typical morphologic and immunophenotypic abnormalities with unremarkable findings by conventional cytogenetics and FISH assays.  This mainly due to the small insertion of the fusion partner gene that cannot be detected by the conventional technique.  The authors reported 4 such cases and reviewed the reported cases in the literature, and then they discussed the techniques that can be used to solve such problem.   The manuscript was well-written and the data were solid and well-presented.  It will be better if the authors can summarize all the techniques discussed into a table  and discuss the advantages and disadvantages of each technique.  This will help the readers to quickly learn which technique could be used in their clinical practices. 

Author Response

 The Cryptic APL is determined by typical morphologic and immunophenotypic abnormalities with unremarkable findings by conventional cytogenetics and FISH assays.  This mainly due to the small insertion of the fusion partner gene that cannot be detected by the conventional technique.  The authors reported 4 such cases and reviewed the reported cases in the literature, and then they discussed the techniques that can be used to solve such problem.  The manuscript was well-written and the data were solid and well-presented.  It will be better if the authors can summarize all the techniques discussed into a table  and discuss the advantages and disadvantages of each technique.  This will help the readers to quickly learn which technique could be used in their clinical practices. 

There are several articles with such configuration. Thus, the author team decided not to duplicate the work. Instead, we added more information for each technique, with advantages and disadvantages, in the discussion section.